# Loss of the Immunomodulatory Transcription Factor BATF2 in Humans Is Associated with a Neurological Phenotype

**DOI:** 10.3390/cells12020227

**Published:** 2023-01-05

**Authors:** Gábor Zsurka, Maximilian L. T. Appel, Maximilian Nastaly, Kerstin Hallmann, Niels Hansen, Daniel Nass, Tobias Baumgartner, Rainer Surges, Gunther Hartmann, Eva Bartok, Wolfram S. Kunz

**Affiliations:** 1Institute of Experimental Epileptology and Cognition Research, Medical Faculty, University of Bonn, 53127 Bonn, Germany; 2Department of Epileptology, University Hospital Bonn, 53127 Bonn, Germany; 3Institute of Clinical Chemistry and Clinical Pharmacology, Medical Faculty, University of Bonn, 53127 Bonn, Germany; 4Institute of Experimental Haematology and Transfusion Medicine, Medical Faculty, University of Bonn, 53127 Bonn, Germany; 5Unit of Experimental Immunology, Department of Biomedical Sciences, Institute of Tropical Medicine, 2000 Antwerp, Belgium

**Keywords:** epilepsy, mental retardation, type I interferonopathy, neuroinflammation, transcription factor

## Abstract

Epilepsy and mental retardation are known to be associated with pathogenic mutations in a broad range of genes that are expressed in the brain and have a role in neurodevelopment. Here, we report on a family with three affected individuals whose clinical symptoms closely resemble a neurodevelopmental disorder. Whole-exome sequencing identified a homozygous stop-gain mutation, p.Gln19*, in the *BATF2* gene in the patients. The BATF2 transcription factor is predominantly expressed in macrophages and monocytes and has been reported to modulate AP-1 transcription factor-mediated pro-inflammatory responses. Transcriptome analysis showed altered base-level expression of interferon-stimulated genes in the patients’ blood, typical for type I interferonopathies. Peripheral blood mononuclear cells from all three patients demonstrated elevated responses to innate immune stimuli, which could be reproduced in CRISPR–Cas9-generated *BATF2*^−/−^ human monocytic cell lines. *BATF2* is, therefore, a novel disease-associated gene candidate for severe epilepsy and mental retardation related to dysregulation of immune responses, which underscores the relevance of neuroinflammation for epilepsy.

## 1. Introduction

Genetic disorders presenting with epilepsy and mental retardation are clinically and genetically heterogeneous. Many of the known disease-associated genes are expressed in the brain and play an important role in neurodevelopmental processes [1,2,3,4,5]. However, a rapidly expanding group of autoinflammatory diseases caused by mutations in genes of the innate and adaptive immune systems have also shown prominent CNS involvement [6]. Here, we report a homozygous loss-of-function mutation in the immunomodulatory transcription factor gene *BATF2* in three siblings of a Turkish family suffering from epilepsy and mental retardation.

BATF2 belongs to the family of ATF-like basic leucine zipper (bZIP) transcription factors, comprised of BATF, BATF2, and BATF3 in humans. Two other bZIP proteins, FOS and c-JUN, form the well-known transcription factor AP-1. Similar to FOS, members of the BATF family can form heterodimers with c-JUN [7], but they lack a transcriptional activation domain, leading to the hypothesis that BATF proteins could competitively inhibit AP-1 function [8]. Since then, different complexes of the BATF family proteins with JUN family proteins have been reported to possess distinct transcriptional activator functions, including unique roles in the differentiation of CD8α+ dendritic cells. BATF2 has also been shown to interact with other transcription factors, such as interferon-regulatory factor 1 (IRF1) [9], as well as form heterotrimers with JUNB and IRF4 or IRF8 [10].

Originally described as an interferon-inducible tumor suppressor through inhibition of AP-1 [8], the immunomodulatory role of BATF2 was first demonstrated in knockout mice. In response to *Trypanosoma cruzi* infection, *Batf2*^−/−^ mice showed faster elimination of the parasite and a higher degree of tissue damage due to elevated immune response [11]. These effects were shown to be consequences of reduced suppression of IL-23 production and subsequent activation of IL-23-dependent pathways. In another study, *Batf2*^−/−^ mice presented with spontaneous colitis which was also accompanied by increased IL-23 production [12]. Similarly, infection with *Schistosoma mansoni* resulted in an aggravated fibro-granulomatous inflammation in *Batf2*^−/−^ mice compared to the controls [13]. However, *Batf2*^−/−^ mice showed reduced tissue inflammation and increased survival after *Mycobacterium tuberculosis* or *Listeria monocytogenes* infection [13], providing evidence for an immunomodulatory rather than a solely immunosuppressive role for this protein. Of note, Batf2 was found to be upregulated in all of these infections, suggesting that Batf2 induction differentially regulates immunological response pathways through a feedback mechanism after exposure to infectious agents.

In humans, increased BATF2 expression has been described in the whole blood of individuals who progressed to active tuberculosis disease in comparison to latently infected individuals who remained healthy [14]. However, to date, no hypomorphic variants of BATF2 have been described in human patients, and the effect of BATF2 deficiency has not been characterized in primary human cells. Here, we show that the loss of BATF2 function represents a novel monogenic etiology of a proinflammatory disease with a severe neurodevelopmental phenotype.

## 2. Materials and Methods

### 2.1. Whole-Exome Sequencing

Genomic DNA was isolated from blood by routine techniques. Whole exome sequencing (WES) was carried out on DNA samples of the three affected siblings, as previously described [15], resulting in a mean coverage of 79–101-fold (30-fold coverage for 81–82% and 10-fold coverage for 94–97% of target sequences). Filtering and variant prioritization were performed using the VARBANK database and analysis tools at the Cologne Center for Genomics. In particular, we filtered for high-quality (coverage > 15-fold; phred-scaled quality > 25) and rare homozygous variants (MAF ≤ 0.05 and not more than one homozygous individual, as based on gnomAD [16]) with predicted effects on protein sequence or splicing. To exclude pipeline-related artifacts, we filtered (MAF ≤ 0.01) against variants from in-house WES datasets from 511 epilepsy patients. Variants reported as benign in the ClinVar database were excluded. Due to high homozygosity scores in the patients suggested that both parents originate from the same inbred population, we also filtered for variants embedded in runs of homozygosity. Direct sequencing of purified PCR products was performed by a commercial service (Eurofins, Ebersberg, Germany).

### 2.2. Transcriptome Analysis

Total blood RNA was isolated with the PAXgene system, while RNA was harvested from a THP-1 cell using the RNeasy Mini Kit (Qiagen, Hilden, Germany). Massively parallel 3′-end mRNA sequencing was performed on a HiSeq 2500 instrument (Illumina, San Diego, CA, USA). There was 100 ng of RNA used for library preparation with QuantSeq 3′-mRNA Library Prep (Lexogen, Vienna, Austria), which was performed according to the manufacturer’s protocol. There were 8.9–16.0 million 50-nt or 100-nt single reads obtained. Mapping to the human genome (GRCh38) and determination of transcript counts were performed using the STAR algorithm (Galaxy Version 2.7.2b) [17] on the Galaxy platform [18]. Differential expression of genes was analyzed using the edgeR package (version 3.26.8) [19]. For differential expression analysis of the patients’ family, female patients were compared to both parents, thus, we excluded sex-specific transcripts from the analysis. In total, 18 genes differentially expressed between the patients and their parents were identified in the 2019 data and 57 in the 2021 data. In THP-1 cells, we found the following total numbers of genes differentially expressed between untreated cells and cells stimulated with TLR7/8 ligands: for wild-type cells, there were 2143 in experiment 1, 936 in experiment 2, and 635 in experiment 3; for *BATF2*^−/−^ cells, there were 2781 in experiment 1, 1464 in experiment 2, and 732 in experiment 3. In order to identify affected pathways, we performed the PANTHER overrepresentation test based on the Gene Ontology database (released 1 July 2022) [20]. Heatmaps of z-scores of normalized gene counts were generated using an in-house R script applying the *scale* and *ggplot* functions.

### 2.3. Cell Line Generation, Culture, and Stimulation

PBMCs were prepared from buffy coats by density gradient centrifugation, as previously described [21]. THP-1 cells were electroporated with a plasmid expressing EF1α promoter-driven Cas9-NLS-2A-EGFP and U6-driven guide RNA targeting BATF2 [CGGGTTCCTGTTACCCAGCTC], sorted for eGFP-positive cells, and selected via limited dilution, as previously described [22]. Genotypes were validated by Sanger sequencing (Appendix A). PBMCs and THP-1 cells were stimulated with herring testes DNA (dsDNA; Sigma-Aldrich/Merck, Darmstadt, Germany), 3pdsRNA, generated by in vitro transcription, as previously described [23], 9.2s RNA (Biomers, Ulm, Germany), Pam3CysK4, ultrapure LPS, flagellin, R848, and CpG2216 (all from InvivoGen, Toulouse, France), as indicated. Prior to stimulation, dsDNA and 3pdsRNA were complexed with Lipofectamine 2000 (Invitrogen/Thermo Scientific, Waltham, MA, USA), and 9.2 s RNA was complexed with poly-L-Arginin (Sigma-Aldrich/Merck, Darmstadt, Germany). Cellular supernatants were then harvested for ELISA probing for IFN-α, IFN-β (Hölzel Diagnostika, Cologne, Germany), TNF, IL-12p40, CXCL10, IL-8, IL-6 (BD Biosciences, Franklin Lakes, NJ, USA), and IL-23 (Human IL-23 HTRF Kit, CisBio, Codolet, France). RNA was isolated, as previously described [24].

### 2.4. Cloning, Generation and Transduction of Lentiviruses

*BATF2* was cloned from cDNA generated from human PBMCs into the 3rd generation lentivector pLenti6-EF1α-IRES-EGFP (a derivative of Invitrogen pLenti6, kindly provided by Jonas Doerr, Institute of Reconstructive Neurobiology, University of Bonn) via SalI/NotI fusion. The p.Gln19* BATF2 point mutant was generated using QuikChange^®^ (Agilent, Santa Clara, CA, USA) site-directed mutagenesis. Lentiviral particles were generated, as previously described [25]: the lentivector plasmid and packaging plasmids D8.9 and pMD2.G were transfected into 293HEK cells using calcium phosphate to produce lentiviral vectors. Viral supernatants were concentrated by ultracentrifugation using a Beckman Coulter (Brea, CA, USA) SW32-TI rotor at 21,000 rpm for 2 h. Virus pellets were resuspended in DMEM and added to human THP-1 with 8 µg/mL polybrene for spin transduction at 32 °C for 90 min. To assure similar expression levels, cells were sorted for GFP expression.

### 2.5. Statistical Analysis

Statistical significance was analyzed via a two-way ANOVA with Bonferroni correction. Differential expression of genes in 3′ RNA-seq analysis was considered significant if the false-discovery rate was below 0.05.

## 3. Results

### 3.1. Clinical Phenotypes of Patients

We investigated three siblings of Turkish origin who shared a phenotype consisting of epilepsy of unknown etiology with a spectrum ranging from gelastic seizures, focal seizures with impaired awareness, tonic, atonic, and tonic-clonic seizures (Figure 1A, Appendix A). Facial features were absent. A detailed clinical description of each patient is presented in Appendix A. The onset of epilepsy was between 1.5 and 14 years. Febrile seizures were observed in patient 1 at one year of age. Two of the three siblings experienced status epilepticus. Furthermore, all three patients presented with psychomotor retardation of a moderate to severe degree with onset between 1.5 and 3 years. All three patients were severely cognitively disabled. Neurological examinations revealed gait abnormalities with ataxia in patients 2 and 3 and tetraparesis in patient 3. Moreover, psychiatric examination showed a behavioral disorder with aggression and hyperactivity in patients 1 and 2. Patient 3 also suffered from frequently recurring gastroenteritis. Blood analysis revealed moderate thrombocytopenia in all three patients, ranging from 80 to 128 Gp/L (reference range: 160–385 Gp/L). Patients 2 and 3 were also found to have leukopenia. Patient 1 presented antinuclear antibodies (ANA), but testing for ANA-subtypes remained negative. Blood cytometric analysis provided other indications of immunopathology in these patients, with a complete lack of B-cells in patient 2 and borderline CD4^+^ T cell counts (454/µL, 450–1500/µL reference range) and low natural killer cells (cell count 12.4/µL, reference range 70–480/µL) in patient 3. CRP and IL-6 levels were in the control range, and no relevant abnormalities were diagnosed in the CSF and MRI of these patients, with one exception: CSF protein was slightly elevated in patient 3 (0.53 g/L, reference range 0.1–0.43). An interictal EEG detected multifocal epileptiform potentials in patient 3. 

### 3.2. Homozygous Stop-Gain Mutation in BATF2

Whole exome sequencing of the three affected siblings showed high total sums of runs of homozygosity (ROH) in each of them. Although no consanguinity of the parents has been reported, they both originated from the same small, inbred population. Thus, we focused on identifying potentially pathogenic homozygous mutations in the patients. Within the detected ROHs, we found a single homozygous deletion that was common in all three patients (Appendix A). This deletion affects the olfactory receptor genes OR4C11, OR4P4, and OR4S2 at the chromosomal region 11q11. A similar deletion at the same region is listed as benign in the ClinVar database (https://www.ncbi.nlm.nih.gov/clinvar/variation/147042/, accessed on 21 November 2022), suggesting that the homozygous deletion identified in the patients is unlikely to be causative. A search for rare homozygous single-nucleotide variants revealed in all three patients potentially pathogenic mutations in the following genes: *BATF2*, *UNC93B1*, *EPOR*, *ZNF709*, and *CCDC105* (Figure 1B, Appendix A). Sanger sequencing confirmed the homozygosity of all these mutations in the patients but showed that the father was also homozygous for the mutations in *EPOR*, *ZNF709*, and *CCDC105*, thus excluding these as candidates.

The predicted pathogenicity of the amino-acid change p.Arg210Gln in *UNC93B1* scores low (22.8) with Combined Annotation-Dependent Depletion (CADD, an integrative annotation built from more than 60 genomic features [26]). In the worldwide population, there are four missense sequence variants in UNC93B1 that have higher CADD scores than the variant in our patients and that were found to be homozygous in more than one individual (gnomAD). All these variants, among which is the mutation of the neighboring amino acid Pro209 (p.Pro209Leu), which also shows even better conservation in the phylogeny than Arg210, are assigned as benign in the ClinVar database. Only three *UNC93B1* nonsense variants are listed in ClinVar as being pathogenic. We, therefore, focused on the homozygous loss-of-function mutation in the *BATF2* gene.

The p.Gln19* stop-gain mutation in *BATF2* is located in the second exon of the canonical transcript isoform 1 and introduces a stop codon upstream of the functionally relevant basic leucine-zipper (bZIP) domain (Figure 1C), thus very likely disrupting translation, and resulting in a truncated, non-functional BATF2 protein (CADD score 35). Considerable readthrough is unlikely, since the mutation generates a UAA stop codon, which has the highest fidelity among the three stop codons [27]. No homozygous truncating mutations in the canonical transcript of *BATF2* were described in over 100,000 individuals in the worldwide population (gnomAD [16]). The two other isoforms of BATF2 are not affected by the mutation (Figure 1C). However, these isoforms lack the basic domain and part of the leucine-zipper domain and are thus predicted to be non-functional. Additionally, we specifically searched for variants in genes previously reported to be associated with epilepsy but found neither homozygous nor heterozygous potential candidate variants. 

### 3.3. Loss of BATF2 Leads to a Type I Interferon Signature in Whole Blood Transcriptome

Since BATF2 has been described as a transcription factor, we aimed to obtain an overview of transcriptional changes related to the patients’ condition. Therefore, we compared the whole-blood transcriptomes of the affected siblings to those of their healthy carrier parents at two different time points (Figure 2A, Appendix A). We observed significant upregulation in numerous interferon-stimulated genes (ISG), including *ISG15*, *IFI44L*, *IFI27*, *IFIT1*, and *RSAD2*, in the patients’ blood, which resembled a persistent type I interferon signature as reported for type I interferonopathies [28]. Note that investigating samples at two different time points reduced the probability that transcriptional differences were distorted by a potential latent viral infection.

### 3.4. Altered Innate Immune Response in Patients’ Peripheral Blood Mononuclear Cells and BATF2^−/−^ THP-1 Cells

BATF2 belongs to the basic leucine zipper (bZIP) family of transcription factors and is predominantly expressed in macrophages and monocytes [29], where it controls macrophage activation and participates in the lineage development of CD8α^+^ and CD103^+^ dendritic cells [10,29]. To characterize the effect of BATF2 loss of function (LOF) on human innate immune signaling, we investigated the response of patient PBMCs to agonists of cytosolic innate immune receptors of nucleic acids, including cGAS and RIG-I and Toll-like receptors (TLR) 2, 4, 5, 7/8, and 9, compared to healthy controls. Patient PBMCs responded with elevated IFN-α, TNF, IL-6, and IL-23 to the stimuli for TLR4 and 7/8, as well as an increased response to the activation of the cytosolic nucleic acid receptors RIG-I and cGAS (Figure 2B–E), although these were not significant for all conditions and cytokines, due in part to the limited number of BATF2-LOF patient donors. Likewise, TLR5 activation by flagellin resulted in higher levels of TNF and IL-6 (Figure 2C,D), which did not reach significance. However, the increased response to imidazoquinoline TLR7/8 agonist R848 and the RNA TLR7/8 agonist 9.2 s RNA [30] were particularly striking, with TNF, IL-6, and IL-23 release increasing threefold in PBMCs from BATF2-LOF patients compared to healthy controls. Here, both agonists were tested due to differences in their resulting cytokine profiles [31].

In contrast, the TLR2 response of BATF2-LOF patients to the agonist Pam3CysK4 was decreased, significantly for TNF (Figure 2B), with a similar tendency, although not significant, for IL-6 (Figure 2C) and IL-23 (Figure 2D). While the cytokine profile observed for TLR2 activation is in line with what has been reported for TLR2 stimulation in *Batf2*^−/−^ mice [9], R848 and LPS stimulation differ substantially between human BATF2-LOF patients and murine bone marrow-derived macrophages (BMDM) from *Batf2*^−/−^ mice or after siRNA knockdown [9,32]. However, it also should be noted that R848 only acts as a TLR7 but not a TLR8 agonist in mice, while in humans, it activates both receptors [33], and important differences in human and murine TLR4 signaling have been reported [34]. Moreover, agonists of TLR5, cGAS, and RIG-I have not been tested in murine *Batf2*^−/−^ cells to date.

In order to confirm that the loss of BATF2 leads to an altered response of the innate immune system in a controlled setting, we generated *BATF2*^−/−^ THP-1 cells as a model for human monocytic cells using CRISPR-Cas9 genome editing. Since R848 and 9.2 s RNA stimulation exhibited the strongest phenotype in BATF2-LOF PBMC, we then analyzed transcriptional changes in wild-type (WT) and *BATF2*^−/−^ THP-1 cells upon stimulation with either R848 or 9.2 s RNA, both agonists of TLR7/8, using two different *BATF2*^−/−^ THP-1 clones (Appendix A) in three independent experiments. Using 3′ RNA sequencing, we did not detect a basal type I interferon signature in non-stimulated *BATF2*^−/−^ THP-1 cells under standard culture conditions (Appendix A). However, we could recapitulate the heightened response to R848 and 9.2 sRNA seen in patient PBMCs. In response to TLR7/8 stimulation, *BATF2*^−/−^ THP-1 cells demonstrated an enhanced induction of genes involved in the antiviral response and type I interferon signaling pathways (Figure 3A,B, Appendix A, cf. [22]). Notably, stimulation-induced expression of the mutated BATF2 mRNA was upregulated in *BATF2*^−/−^ THP-1 cells, consistent with its role as an ISG [8]. In line with these findings at the mRNA level, a stimulation with R848 and 9.2 sRNA in *BATF2*^−/−^ THP-1 cells led to an elevated CXCL10 and IL-12p40 response and an increased release of IFN-β, TNF and the proinflammatory cytokines IL-6 and IL-8 (Figure 3C–H). Moreover, *BATF2*^−/−^ THP-1 cells also recapitulated the enhanced cytokine release observed downstream of cGAS and RIG-I activation in patient PBMCs. Moreover, TLR2 stimulation using Pam3CysK4 resulted in a decreased response of TNF, IL-12p40, CXCL10, and IL-8 in *BATF2*^−/−^ THP-1 cells as compared to controls, in line with our findings in patient PBMCs and previous observations in *Batf2*^−/−^ mice [9]. Nevertheless, TLR4 could not be sufficiently examined, likely due to the weak expression of CD14 in THP-1 cells, which results in a dampened response to LPS [35].

To confirm that the altered immune phenotype observed in *BATF2*^−/−^ THP-1 cells is a direct consequence of the lack of functional BATF2 isoform 1, as well as rule out the influence of off-target effects during genome editing, we re-expressed wild-type BATF2 and the p.Gln19* mutant in *BATF2*^−/−^ THP-1 cells (Figure 4). In line with our other data, wild-type BATF2 could reverse the heightened CXCL10 response to TLR4, TLR7/8, cGAS, and RIG-I stimulation, while p.Gln19* mutant expression was without effect. Moreover, wild-type BATF2 restored CXCL10 expression upon TLR2 activation.

## 4. Discussion

Rare genetic disorders of autoinflammatory and autoimmune origin that are characterized by elevated activity of the type I interferon pathway have been recently recognized as a distinct pathological entity and are referred to as type I interferonopathies [36,37]. A common feature of this type of disease is an increased basal expression of interferon-stimulated genes in the blood of affected patients. We demonstrate here that three siblings suffering from epilepsy and mental retardation due to a homozygous loss-of-function mutation in *BATF2* show important hallmarks of type I interferonopathy.

The constitutive upregulation of type I interferon is common to a group of monogenic diseases known as Aicardi–Goutières syndrome (AGS), which is characterized by leukoencephalopathy with basal ganglia calcification, accompanied by dystonia and seizures with onset in early infancy [38]. Some patients also show the presence of anti-nuclear antibodies and develop arthritis, thrombocytopenia, and lymphopenia, which are also typical symptoms of systemic lupus erythematosus (SLE), an autoimmune disorder that has also been linked to inappropriate type-I interferon release. Although no structural alterations of the brain have been reported in the three patients described in this study, they suffered from prominent neurological symptoms, such as seizures of different types, psychomotor retardation, and cognitive disability. Furthermore, they also presented with minor clinical features indicative of an autoinflammatory disorder, such as discrete leukopenia (with a lack of B cells in patient 2 and low CD4^+^ T cells and natural killer cells in patient 3), thrombocytopenia, and the presence of anti-nuclear antibodies. 

Type I interferonopathies are closely linked to the heightened activation of nucleic acid sensing and the innate immune antiviral response [39]. Known causes include hypermorphic variants of nucleic acid receptors, such as MDA-5, RIG-I, and STING, and hypomorphic variants of negative regulators of nucleic acid sensing and the antiviral response, including nucleases, such as TREX1 [40], and inhibitors of the type I interferon pathway, such as USP18 [38]. A role for the *BATF2* gene in autoinflammatory diseases and type I interferonopathy has not been reported to date. However, the presence of a type I interferon signature in the basal transcriptome of BATF2-deficient patients suggests a regulatory role of BATF2 in the type I interferon response, as does the heightened response of *BATF2*^−/−^ THP-1 cells to nucleic acid stimuli. The apparently contradictory observation that we were not able to detect, such a basal type I interferon signature in *BATF2*^−/−^ THP-1 cells, is likely due to highly controlled, stimulus-free culturing conditions, which differ substantially from the environment of human organisms. Thus, we interpret the type I interferon signature in human patients as a chronically activated response to long-term, recurrent exposure to cellular stress and pathogens [41,42]. 

In mice, it has been shown that Batf2 has a nonredundant role in the regulation of the AP 1-dependent inflammatory response. During *Trypanosoma cruzi* infection, *Batf2*^−/−^ mice demonstrated an elevated level of the AP-1 cytokine IL-23, accompanied by an enhanced Th17 immune response, which was reverted in *Batf2*^−/−^ *Il23a*^−/−^ animals [11]. Similarly, another study demonstrated that spontaneous colitis in *Batf2*^−/−^ mice was abrogated by additional Il-23 deficiency [12]. However, in in vitro murine BMDMs, expression of IL-12p40, a subunit of IL-23, was decreased after BATF2 depletion using siRNA and stimulation with the TLR4 ligand LPS [9], and another study, using *Batf2*^−/−^ BMDMs, demonstrated this for specific stimulation of the TLR7 pathway by R848 [32]. In contrast, our data demonstrates enhanced production of IL-23 in patient PBMCs after stimulation with TLR4 and TLR7/8 agonists, as well as increased IL-12p40 release after TLR7/8 stimulation in *BATF2*^−/−^ THP-1 cells. These data seem to better reflect murine in vivo IL-23 release downstream of *T. cruzi* infection, an immune stimulus that likely activates multiple PRRs. There are several possible explanations for this difference, including the cell types investigated (BMDMs vs. PBMCs) and key differences in human and murine PRR activation. While R848 solely activates the TLR7 pathway in mice, in humans, it activates both TLR7 and TLR8 [33], and the IL-12p40/IL23 axis is primarily driven by TLR8, not TLR7, in human PBMC [21,43,44]. In addition, important differences between mice and humans in the function of TLR4 and its accessory protein MD2 have been described [34,45,46,47].

In contrast to the patients in our study, *Batf2*^−/−^ mice do not have a detectable neurological phenotype. However, it should be noted that many of the AGS-related mouse models lack the pronounced neurological defect found in many AGS patients, while nonetheless retaining proinflammatory hallmarks in other organs [48]. Importantly, one of the three patients suffered from recurrent gastroenteritis which resembles the phenotype of *Batf2*^−/−^ mice [12].

Activation of the innate immune system, such as the application of the TLR4 agonist LPS, is known to be potently epileptogenic in mouse models, and chronic inflammation has been implicated in mesial temporal lobe epilepsy with hippocampal sclerosis, a frequently occurring, acquired type of epilepsy [49]. This activation of the innate immune system does not only lead to the release of epileptogenic proinflammatory cytokines but also the recruitment of cells of the adaptive immune system into the CNS. Previously, we reported increased numbers of T lymphocytes in the hippocampal parenchyma of patients with hippocampal sclerosis, which were accompanied by signs of mitochondrial DNA damage, putatively of oxidative origin [50]. Damaged mitochondrial DNA is now recognized as a potential trigger of type I interferon-dependent antiviral innate immune response that acts through the activation of the cGAS–STING pathway [51]. This still hypothetical overlap between genetic and acquired forms of inflammation-related epilepsy underscores the relevance of neuroinflammatory processes in the generation of seizures and harbors great potential for the development of novel therapeutic approaches. 

## 5. Conclusions

We demonstrate that the loss of the BATF2 transcription factor leads to a multifaceted dysregulation of inflammatory responses, alters expression of interferon-stimulated genes, consistent with type I interferonopathy, and represents a novel candidate gene associated with a proinflammatory disease with a severe neurological phenotype. 

## Figures and Tables

**Figure 1 cells-12-00227-f001:**
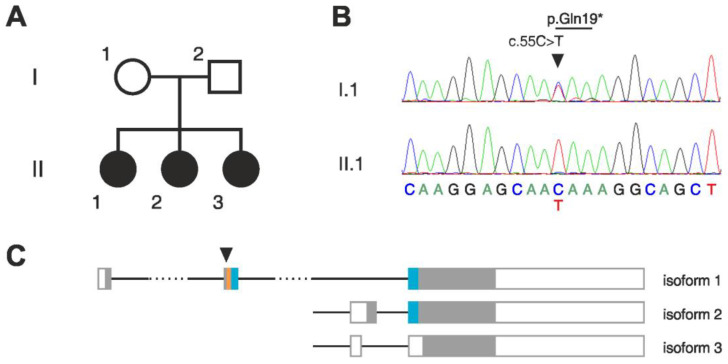
Homozygous stop-gain mutation in *BATF2* in a family with three affected children. (**A**) Pedigree of the affected family. (**B**) Sequencing chromatograms of isoform 1 cDNA showing the *BATF2* p.Gln19* mutation in heterozygous state in the mother (I.1) and in homozygous state in the index patient (II.1). (**C**) Structures of the three transcript isoforms of *BATF2*. Filled boxes, protein coding region; empty boxes, non-coding region; orange, basic region; blue, leucine-zipper domain; arrowhead, position of the mutated nucleotide. Note that only isoform 1 contains the complete basic-leucine zipper (bZIP) domain which is required for DNA binding and dimerization.

**Figure 2 cells-12-00227-f002:**
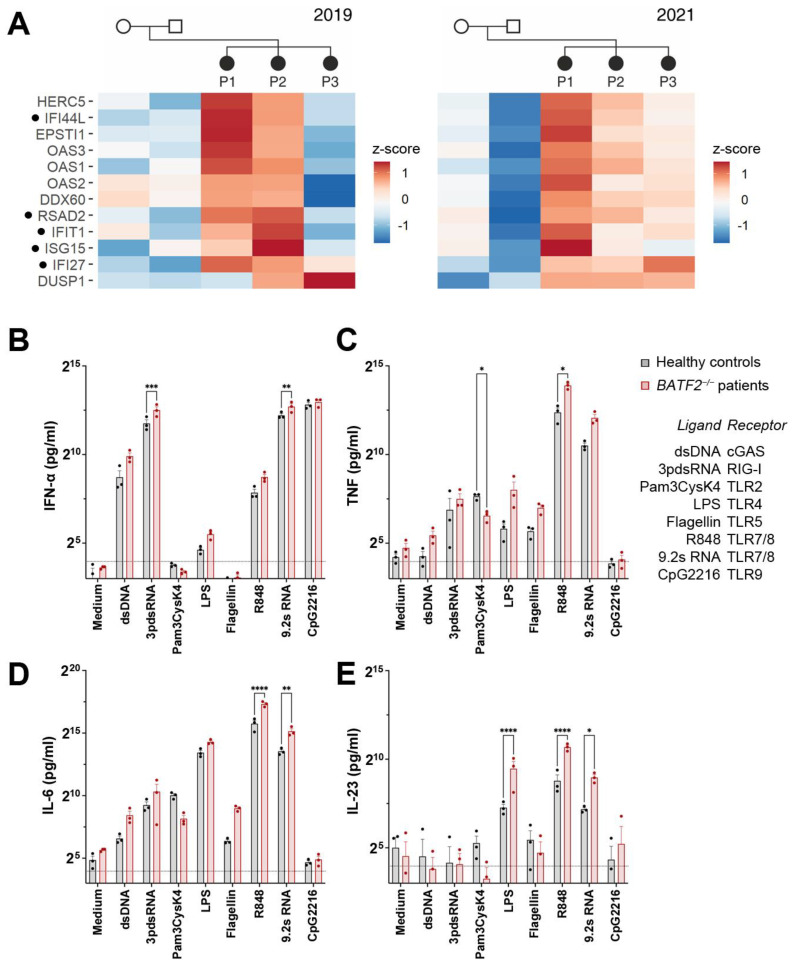
Loss of BATF2 causes elevated expression of interferon-stimulated genes in patients. (**A**) Differentially upregulated genes in the patients’ whole-blood mRNA as compared to their unaffected parents by 3′ RNA sequencing. Heatmaps of z-scores of normalized gene counts are shown from two different time points (years 2019 and 2021). Genes are shown that were significantly upregulated (FDR < 0.05) in patients at least at one of the time points and always displayed higher values in at least two of the patients compared to the parents. Genes associated with type I interferonopathies [28] are indicated by dots. P1–3, patients. NB: patient 3 has severe leukopenia, which might affect the detected expression of interferon-stimulated genes. (**B**–**E**) In-vitro TLR7/8 stimulation of peripheral blood mononuclear cells (PBMCs) obtained from the *BATF2* p.Gln19* patients demonstrate upregulated cytokine release. PBMCs were stimulated with double-stranded DNA (ligand of cGAS), double-stranded RNA (ligand of RIG-I), 200 pg/mL Pam3CysK4 (ligand of TLR2), 100 pg/mL LPS (ligand of TLR4), flagellin (ligand of TLR5), 10 µM R848 or 9.2 s RNA (ligand of TLR7/8), or CpG DNA (ligand of TLR9). The corresponding receptors are indicated in the figure legend. Cellular supernatants were harvested for IFN-α, TNF, IL-6, and IL-23 ELISA. Arithmetic means ± SEM from three independent experiments is presented. Statistical significance was analyzed via a two-way ANOVA with Šidák’s correction. *, *p* < 0.05; **, *p* < 0.01; ***, *p* < 0.005; ****, *p* < 0.001.

**Figure 3 cells-12-00227-f003:**
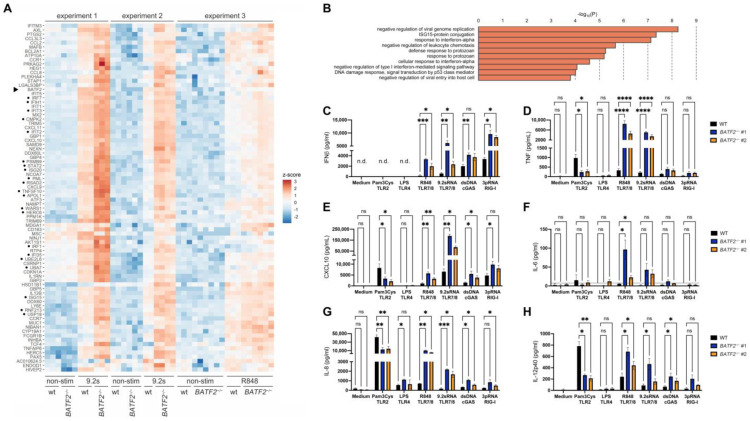
Knockout of *BATF2* leads to elevated TLR7/8 stimulation and repressed TLR2 response in THP-1 cells. (**A**) Differentially expressed genes in *BATF2*^−/−^ and wild-type THP-1 cells identified by 3′ RNA sequencing. Heatmap of z-scores of normalized transcript counts is shown for genes that were significantly upregulated upon TLR7/8 stimulation in *BATF2*^−/−^ cells (FDR < 0.05) and displayed at least 2-fold higher upregulation in *BATF2*^−/−^ in comparison to wild-type cells in at least two of the three independent experiments. *BATF2*^−/−^ and wild-type (WT) THP-1 cells were stimulated with 2 µg/mL R848 or 7.5 µg/mL 9.2 s RNA. Genes associated with type I interferonopathies [28] are indicated by dots. Arrowhead, BATF2. (**B**) Gene ontology pathway enrichment analysis of upregulated genes. Only pathways with at least 30-fold enrichment and more than two hits are shown. (**C**–**H**) Cytokine release is upregulated upon TLR7/8 and cGAS stimulation and downregulated upon TLR2 stimulation in *BATF2*^−/−^ THP-1 cells. *BATF2*^−/−^ and wild-type (WT) THP-1 cells were stimulated with 2 µg/mL Pam3Cys, 2 µg/mL LPS, 500 ng/mL R848, 2 µg/mL 9.2 s RNA, 250 ng/mL dsDNA, or 100 ng/mL 3pdsRNA. Cellular supernatants were harvested for ELISA measuring IFN-β (coded by gene *IFNB1*) (**C**), TNF (**D**), CXCL10 (**E**), IL-6 (**F**), IL-8 (coded by gene *CXCL8*) (**G**), or IL-12p40 (coded by gene *IL12B*) (**H**). The dotted line denotes the detection limit of the ELISA used. Arithmetic mean ± SEM from three independent experiments is presented. Statistical significance was analyzed via two-way ANOVA with Dunnett’s correction. *, *p* < 0.05; **, *p* < 0.01; ***, *p* < 0.005; ****, *p* < 0.001.

**Figure 4 cells-12-00227-f004:**
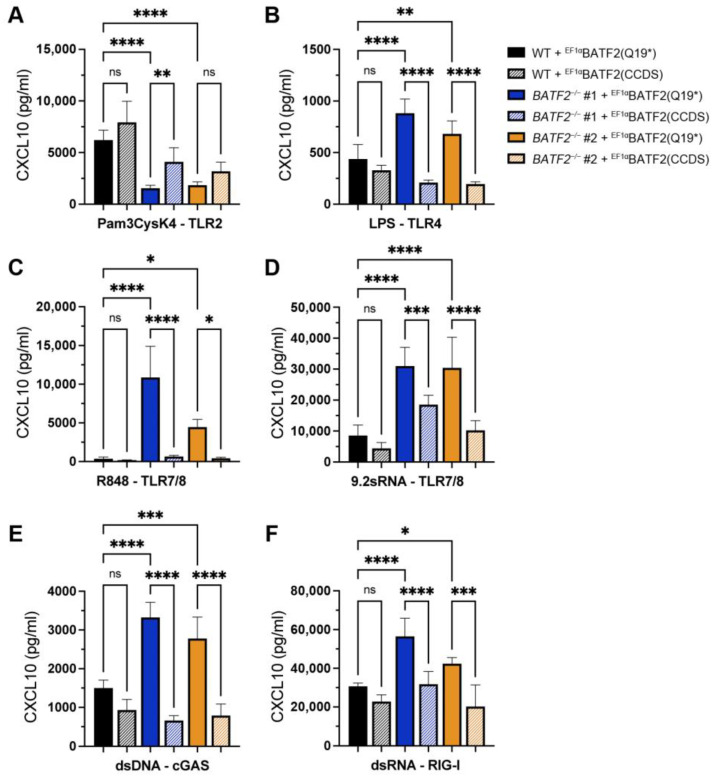
Rescue of TLR7/8, cGAS, and RIG-I responses by lentiviral expression of wild-type BATF2. Wild-type and *BATF2*^−/−^ THP-1 cells ectopically expressing wild-type or p.Gln19* mutant BATF2 were stimulated with 2 µg/mL Pam3Cys (**A**), 2 µg/mL LPS (**B**), 500 ng/mL R848 (**C**), 2 µg/mL 9.2 sRNA (**D**), 250 ng/mL dsDNA (**E**), or 100 ng/mL 3pdsRNA (**F**). Cellular supernatants were harvested for CXCL10 ELISA. Arithmetic mean ± SEM from three independent experiments is presented. Statistical significance was analyzed via a one-way ANOVA with Šidák’s correction. *, *p* < 0.05; **, *p* < 0.01; ***, *p* < 0.005; ****, *p* < 0.001.

## Data Availability

RNA-seq data on THP-1 cells are available in the GEO database under accession number GSE220916. Other datasets generated during the current study are available from the corresponding authors upon reasonable request.

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
