# Peer review of "Loss of the Immunomodulatory Transcription Factor BATF2 in Humans Is Associated with a Neurological Phenotype"

_cells, 2023, doi:10.3390/cells12020227_

Round 1

Reviewer 1 Report

Zsurka et al report on a novel candidate gene associated with neurological phenotype in humans.

Using WES they revealed a homozygous stop-gain 22 mutation p.Q19* in the BATF2 gene in the patients. They showed that Peripheral blood mononuclear cells from all three patients demonstrated elevated responses to innate immune stimuli, which could be reproduced in CRISPR–Cas9-generated BATF2-/- human monocytic cell lines.

Data is very interesting, would improve if taken care of few minor concerns.

1.       This is the first family that have been associated with BATF2 associated neurological phenotype in humans. I would recommend considering it as a novel candidate gene. Require several other families to confirm.

2.       Follow HGVS nomenclature: https://varnomen.hgvs.org/

3.       Three letter abbreviations for amino acids are mostly used.

4.       Why MAF ≤ 0.01 was used?

5.       IRB approval number should be added.

6.       Was consent obtained to publish patient data?

7.       Non-consanguineous family? According to pedigree?

8.       Clinical description is weak: Better to explain each patient separately.

9.       MRI, CT, facial features?

10.   Future of the family? Any medications advised?

11.   Comment on Prenatal genetic diagnosis, NIPT, PGT-M, new born screening: PMID: 36406136, PMID: 33804821, PMID: 33613643, PMID: 31557427.

12.   References that could be cited: PMID: 34415064, PMID: 34702355 , PMID: 32246862, PMID: 34820905, PMID: 31852446.

Author Response

Reviewer 1:

Zsurka et al report on a novel candidate gene associated with neurological phenotype in humans.

Using WES they revealed a homozygous stop-gain 22 mutation p.Q19* in the BATF2 gene in the patients. They showed that Peripheral blood mononuclear cells from all three patients demonstrated elevated responses to innate immune stimuli, which could be reproduced in CRISPR–Cas9-generated BATF2-/- human monocytic cell lines.

Data is very interesting, would improve if taken care of few minor concerns.

  1. This is the first family that have been associated with BATF2 associated neurological phenotype in humans. I would recommend considering it as a novel candidate gene. Require several other families to confirm.

: We corrected the abstract and the conclusion to candidate gene.

  1. Follow HGVS nomenclature: https://varnomen.hgvs.org/

: Done.

  1. Three letter abbreviations for amino acids are mostly used.

: We changed p.Q19* to p.Gln19* throughout the text.

  1. Why MAF ≤ 0.01 was used?

: MAF≤ 0.05 was used for the general population, MAF ≤ 0.01 for the in-house database

  1. IRB approval number should be added.

: The IRB approval number is listed in the paragraph ‘Institutional Review Board Statement’ at the very end of the manuscript.

  1. Was consent obtained to publish patient data?

: Yes, written informed consent was obtained to publish the video EEG monitoring and all other patient data in an anonymous form. (cf. Informed Consent Statement at the end of the manuscript)

  1. Non-consanguineous family? According to pedigree?

: The parents reported no consanguinity, but originate from the same small village in Turkey.

  1. Clinical description is weak: Better to explain each patient separately.

: A detailed description of each patient is in Supplementary Table S1.

  1. MRI, CT, facial features?

: This information is in Suppl. Tab. S1. Facial feature were absent. That has been added to paragraph 3.1.

  1. Future of the family? Any medications advised?

: Present medication is provided in Supplementary Table S1.

  1. Comment on Prenatal genetic diagnosis, NIPT (noninvasive prenatal test), PGT-M, new born screening: PMID: 36406136, PMID: 33804821, PMID: 33613643, PMID:31557427.

: This is extremely rare disorder due to the small protein size (274 AS) and recessive trait. A prenatal testing is therefore from our point of view not appropriate.

  1. References that could be cited: PMID: 34415064 (biallelic PUS 3=, PMID: 34702355 (de novo CACNA1E), PMID: 32246862 (homozygous NKX6-2), PMID: 34820905 (homozygous PPP1R1B/DARPP-32), PMID: 31852446 (homozygous HEXB and MBOAT7).

: Included in the manuscript

Reviewer 2 Report

In this manuscript Kunz and collaborators associated a pathogenic variant of BATF2 gene with a neurological phenotype, closely resemble to a neurodevelopmental disorder. This manuscript is well written, and the study design has been very well conducted. 

The authors identified a stop-gain mutation (p.Q19*) in BATF2 gene performing the whole-exome sequencing in three Turkey sibling, both affected by neurological disorder, and associated this pathogenic variant to the symptoms of the patients. The authors also analyzed the whole-blood transcriptome of the three patients and investigated their innate immune response to different stimuli. Moreover, to corroborate their hypothesis, they generated mutant human monocytic cells mutated in BATF2 gene using CRISPR-Cas9 technology. 

Although the manuscript is well written and experiment are well conducted, there are some critical points that I’d like authors address before publication. 

Major points: 

One of my major concerns is about the lack of NGS data informations in the manuscript. The authors should improve Material and Methods section adding additional information about the analysis of Transcriptome data. This will help the reader to understand how analysis have been done. 

In the section 2.2 -“Transcriptome analysis”- the authors should add more information about transcriptomic analysis:

i)               Which are the characteristic of the sequencing?  ; How deep was the sequencing? (i.e. how many reads (in millions) they obtained from sequencing?). 

ii)              The authors should reported also statistical data about how many reads passed quality check (if it has been done) 

iii)            Which genomic release have been used to map reads? (GRCh38?)

iv)            How many reads uniquely mapped on the reference genome?

v)              Which version of STAR and EdgeR has been used for the analysis?

vi)            How many genes the authors found Differentially expressed in the different analysis that they conducted?

vii)           How the authors generated heatmaps of DE genes? 

They should add this information for both whole-blood and WT/Mutant cellular transcriptome.

Moreover, the reviewer think that is very important that the authors should provide in the manuscript

i)               supplementary files all the DE found in their analysis and the Gene-ontology analysis. This will help the reader. 

ii)             Accession Number of row-data. I think that is important, unless a specific motivation, that all the data generated for this research should be deposited in an open access database, such ad GEO database. Accession number should be included in the specific manuscript section (Data Availability Statement).

How the authors generated BATF2-/- THP-1 cells? I think that Material and Methods section lacks some important information about the generation of BATF2-/- using CRISPR-Cas9 technologyWhich strategy has been followed for CRISPR-Cas9  Knock-out? How the authors evaluated the possibility of off-target?

Minor points:

-Introduction: The authors could discuss in the introduction section that BATF2 gene encoding for three different isoforms and what is known about their role. This will help the redear to better understand results section. 

-it is not clear why the authors decided to perform whole-blood transcriptome analysis at two different time point. The authors should discuss this point in the result section. 

- p.Q19* in BATF2 gene cause a premature stop-codon that doesn’t allow the formation of the isoform 1 of BATF2. Could be very informative that authors demonstrate the lack of isoform 1 of BATF2 protein in the three patientsmaybe performing wester-blot analysis in patients-derived PBMC cells. 

Author Response

Reviewer 2:

In this manuscript Kunz and collaborators associated a pathogenic variant of BATF2 gene with a neurological phenotype, closely resemble to a neurodevelopmental disorder. This manuscript is well written, and the study design has been very well conducted. 

The authors identified a stop-gain mutation (p.Q19*) in BATF2 gene performing the whole-exome sequencing in three Turkey sibling, both affected by neurological disorder, and associated this pathogenic variant to the symptoms of the patients. The authors also analyzed the whole-blood transcriptome of the three patients and investigated their innate immune response to different stimuli. Moreover, to corroborate their hypothesis, they generated mutant human monocytic cells mutated in BATF2 gene using CRISPR-Cas9 technology. 

Although the manuscript is well written and experiment are well conducted, there are some critical points that I’d like authors address before publication. 

Major points: 

One of my major concerns is about the lack of NGS data informations in the manuscript. The authors should improve Material and Methods section adding additional information about the analysis of Transcriptome data. This will help the reader to understand how analysis have been done. 

In the section 2.2 -“Transcriptome analysis”- the authors should add more information about transcriptomic analysis:

  1. i)Which are the characteristic of the sequencing?  ; How deep was the sequencing? (i.e. how many reads (in millions) they obtained from sequencing?). 
  2. ii)The authors should reported also statistical data about how many reads passed quality check (if it has been done) 

iii)            Which genomic release have been used to map reads? (GRCh38?)

  1. iv)How many reads uniquely mapped on the reference genome?
  2. v)Which version of STAR and EdgeR has been used for the analysis?
  3. vi)How many genes the authors found Differentially expressed in the different analysis that they conducted?

vii)           How the authors generated heatmaps of DE genes?They should add this information for both whole-blood and WT/Mutant cellular transcriptome.

: All these information is now included in the corresponding paragraph of Methods.

Moreover, the reviewer think that is very important that the authors should provide in the manuscript

  1. i)supplementary files all the DE found in their analysis and the Gene-ontology analysis. This will help the reader.

: Three new Excel files (Files S1 – S3) are now included in the Supplementary Materials.

  1. ii)Accession Number of row-data. I think that is important, unless a specific motivation, that all the data generated for this research should be deposited in an open access database, such ad GEO database. Accession number should be included in the specific manuscript section (Data Availability Statement).

: RNA-seq data on THP-1 cells has been submitted to the GEO database and will be available after publication under the accession number GSE220916. The GEO database does not accept NGS data on human individuals due to privacy protection.

How the authors generated BATF2-/- THP-1 cells? I think that Material and Methods section lacks some important information about the generation of BATF2-/- using CRISPR-Cas9 technologyWhich strategy has been followed for CRISPR-Cas9  Knock-out? How the authors evaluated the possibility of off-target?

: Generation of the CRISPR-Cas9 knockout cells lines is described in Section 2.3. The facts that we performed experiments on two independent BATF2-/- cell lines and that the observed effects were reversed by the expression of wild-type, but not mutant BATF2 makes it very unlikely that the observed effects were due to an off-target genomic alteration.

Minor points:

-Introduction: The authors could discuss in the introduction section that BATF2 gene encoding for three different isoforms and what is known about their role. This will help the redear to better understand results section. 

: There is not literature on isoforms 2 and 3, because they are lacking the functional domains for DNA binding. This has been outlined in the results section (end of paragraph 3.2).

-it is not clear why the authors decided to perform whole-blood transcriptome analysis at two different time point. The authors should discuss this point in the result section. 

: Transcriptional changes in immunologically relevant genes could theoretically be caused by latent infections in the patients. By obtaining blood samples at two different time points reduces the probability of this artifact significantly. This is now stated in in Section 3.3.

- p.Q19* in BATF2 gene cause a premature stop-codon that doesn’t allow the formation of the isoform 1 of BATF2. Could be very informative that authors demonstrate the lack of isoform 1 of BATF2 protein in the three patientsmaybe performing wester-blot analysis in patients-derived PBMC cells. 

: Unfortunately, this is due to the extremely low levels of this transcription factor not feasible. The expression ratio of isoform 1 to GAPDH is in unstimulated cells 0.0002.

Round 2

Reviewer 2 Report

Thanks to the authors that addressed all my concerns